# The Influence of Inter-Enterprise Knowledge Heterogeneity on Exploratory and Exploitative Innovation Performance: The Moderating Role of Trust and Contract

**Chen Tao [1], Yiying Qu [2],\*, Hao Ren [1] and Zhuopin Guo [1]**

[1]   School of Economics and Management, Tongji University, Shanghai 200092, China; afratao@126.com (C.T.); renhao_cn@hotmail.com (H.R.); guozhuopin@126.com (Z.G.)

[2]   Business School, East China University of Political Science and Law, Shanghai 200042, China

\*   Correspondence: sdlzqyy@hotmail.com

**Abstract:** Improving enterprise innovation performance is key for enterprises to obtain sustainable competitiveness. With the increasingly fierce market competition of technological and product innovation, acquiring external heterogeneous knowledge of alliance enterprises becomes core to improving innovation performance. In this paper, we constructed a theoretical model to present the effect of inter-enterprise knowledge heterogeneity and alliance network governance mechanisms on enterprise innovation performance. We selected high-tech enterprises as the research object for empirical research and reached the following conclusions: (1) Inter-enterprise knowledge heterogeneity has a positive effect on exploratory and exploitative innovation performance, and (2) trust and contract have a moderating effect on the relationship between inter-enterprise knowledge heterogeneity and enterprise innovation performance.

**Keywords:** inter-enterprise knowledge heterogeneity; exploratory innovation performance; exploitative innovation performance; trust; contract

---

## 1. Introduction

Technological innovation ability is a key factor in acquiring sustained competitiveness for an enterprise. On the one hand, the realization of enterprise innovation needs effective access to acquire external heterogeneous knowledge. On the other hand, it needs contingency governance mechanisms of alliance networks to realize an effective absorption and integration of heterogeneous knowledge, thereby improving enterprise innovation performance. The influence of inter-firm heterogeneous knowledge and the R&D alliance governance mechanism on enterprise innovation performance has attracted the attention of many scholars.

Some scholars proposed that heterogeneous knowledge can promote enterprise innovation performance due to the scale economy and sharing effect (Subramaniam, 2005; Suzuki, 2004) [1,2]. However, some scholars believe that heterogeneous knowledge negatively affects innovation because it easily produces conflicts and is difficult to transfer and absorb (Tanriverdi & Venkatraman, 2005; Lin, 2005) [3,4]. Considering the benefits and costs of heterogeneous knowledge on innovation, some scholars have verified the non-linear relationship between heterogeneous knowledge and innovation with an inverted U function (Sampson, 2007; Oerlemans, 2013; Ye J.F., Hao B., etc., 2016) [5–7].

Although the R&D alliances provide a way for enterprises to acquire heterogeneous knowledge from alliance partners, they do not guarantee that enterprises can effectively identify, share and absorb heterogeneous knowledge from alliance partners. The increasing heterogeneity of knowledge will

aggravate this problem, because the relative absorption capacity of enterprises will decline (Lane & Lubatkin, 1998) [8]. High heterogeneity reduces the opportunities for alliance members to share common understanding, common language and thinking patterns (Cohen & Levinthal, 1990) [9].

Therefore, the alliance governance mechanism, as an important means, plays a moderating role between heterogeneous knowledge and innovation performance (Phelps, 2010) [10].

Considering the previous literature on enterprise innovation performance, we identify two deficiencies. First, few studies investigate the influence of inter-enterprise knowledge heterogeneity on exploratory and exploitative innovation performance separately, and most of the studies are general studies on the relationship between knowledge acquisition and enterprise innovation performance. Second, research conclusions are seriously lacking about the moderating effects of the formal governance mechanism (contract) and informal governance mechanism (trust) on the relationships between inter-enterprise knowledge heterogeneity and exploratory and exploitative innovation performance in R&D alliances.

Therefore, based on a background of network cooperation and innovation of R&D alliances between enterprises, this study proposes the following scientific questions: (1) Is there a relationship between inter-enterprise knowledge heterogeneity, exploratory innovation and exploitative innovation? (2) Do trust and contract have a moderating effect on the relationship between inter-enterprise knowledge heterogeneity and enterprise innovation performance?

To solve these research questions, we constructed the conceptual model based on a broad review of the literature and case studies. Then, a theoretical analysis of the research questions was conducted, and relevant hypotheses were tested. Finally, we obtained 322 samples of high-tech enterprises in the city of Shanghai and the provinces of Jiangsu, Zhejiang, Anhui and Guangdong; used the hierarchical regression analysis method for empirical research; and obtained important results. Conclusions are as follows. (1) Inter-enterprise knowledge heterogeneity has a positive effect on exploratory innovation and exploitative innovation performance. (2) Trust and contract have a moderating effect on the relationship between inter-enterprise knowledge heterogeneity and enterprise innovation performance.

After the introduction, we sort the relevant theories and put forward the hypothesis in Section 2, introduce the relevant methods in Section 3, present the empirical results in Section 4 and discuss the research conclusions in Section 5. This paper enriches the existing theoretical achievements on the relationship between inter-enterprise knowledge heterogeneity and exploratory and exploitative innovation performance, and fills the research gaps on the relationship between trust and contract-moderated knowledge heterogeneity and enterprise innovation performance. All these contributions are presented in Section 5.

## 2. Theoretical Foundations and Hypothesis Development

### 2.1. Inter-Enterprise Knowledge Heterogeneity and Innovation Performance

Wernerfelt (1984) [11] formally proposed a resource-based view of the firm, wherein he held that the internal resources of an enterprise play a more decisive role in obtaining a competitive advantage than the external environment does, and the accumulation of enterprise resources, knowledge and organization ability is the key to maintaining its competitive advantage. The knowledge-based view holds that knowledge is the most strategically significant resource of an enterprise (Grant, 1996) [12], and the creation, transfer, storage and application of heterogeneous knowledge could generate new knowledge with economic value and bring a sustainable competitive advantage.

With the increasing differences in technical knowledge among enterprises, their ability to identify, absorb and apply knowledge to each other is declining. Cohen & Levinthal (1990) [9] put forward the theoretical view of absorptive capacity, while knowledge and network research scholars were expanding the traditional resource-based view. A high level of absorptive capacity can push enterprises to make better use of other enterprises' heterogeneous technology and knowledge, thereby motivating their innovation activities.

When the degree of knowledge heterogeneity exceeds a certain limit, the lack of cognitive ability and experience in absorbing heterogeneous knowledge will limit enterprises' ability to understand the interaction between heterogeneous complex elements (Vrande and Vareska., 2013) [13]. Integrating novel knowledge from heterogeneous resources also usually requires changing the existing communication mode and social exchange mode, which is difficult to implement in established organizations (Ye J.F., Hao B., etc., 2016) [7]. When the degree of inter-enterprise knowledge heterogeneity exceeds a specific threshold, the cost of absorbing and utilizing knowledge is also greatly increased, negatively affecting innovation performance.

Phelps (2010) [10] studies the relationship between network diversity and exploratory innovation. He believes that network diversity increases the novelty of the knowledge acquired by enterprises from the network, affecting exploratory innovation. Considering scholars' different conclusions on the relationship between knowledge heterogeneity and enterprise innovation performance, this study attempts to subdivide and consider enterprise innovation performance and explore the influence of knowledge heterogeneity on two different types of innovation, namely, exploratory and exploitative innovation.

Exploratory innovation is the result of finding new organizational practices and discovering new methods, technologies, businesses, processes and products (Lin, McDonough, 2014) [14]. Its goal is to meet the needs of emerging customers and markets by providing new designs, creating new products/services and developing new sales channels (Li, Lin, et al., 2014) [15].

Exploitative innovation is based on existing technology, customer and market knowledge and existing skills and processes (Lin, Chang, 2015) [16].

In this study, the degree of inter-enterprise knowledge heterogeneity provides enterprises with diversified knowledge and information different from their own knowledge fields. Given that exploratory innovation is the technological innovation that needs to acquire and create brand new knowledge and to break away from, and surpass, the existing knowledge base, the degree of inter-enterprise knowledge heterogeneity can influence the performance of exploratory technological innovation.

Exploitative innovation is mainly the technological innovation based on existing knowledge. Therefore, the emphasis is placed on refining, integrating, strengthening and improving the existing knowledge, which tends to integrate and use similar knowledge among enterprises. The high level of inter-enterprise knowledge heterogeneity does not lead enterprises to focus on the integration and improvement of their own internal knowledge. Therefore, this study holds that the degree of inter-enterprise knowledge heterogeneity negatively affects the performance of exploitative innovation.

Based on the theoretical analysis, we proposed the following hypotheses.

**Hypothesis 1 (H1).** *Inter-enterprise knowledge heterogeneity has a positive effect on the exploratory innovation performance of an enterprise.*

**Hypothesis 2 (H2).** *Inter-enterprise knowledge heterogeneity has a negative effect on the exploitative innovation performance of an enterprise.*

## 2.2. Moderating Role of Trust

Although the model of inter-enterprise alliance, as regards scientific research activities, provides enterprises with access to the heterogeneous knowledge of alliance partners, it does not guarantee that enterprises can effectively identify, share and absorb them (Hamel, 1991) [17]. The tacit and embedded nature of technical knowledge makes it difficult to be recognized, spread and absorbed by member partners (Teece, 1992) [18], thereby reducing the possibility of a successful combination (Galunic and Rodan, 1998) [19].

Trust as an informal organizational mechanism, was a critical supplement to formal mechanisms, and trust governance plays an increasingly important role in inhibiting opportunism and promoting inter-firm cooperation (Dyer and Singh, 1998) [20].

Trust among enterprises will reduce the extent of knowledge protection and enhance knowledge sharing and innovation (Kale et al., 2000; Larson, 1992) [21,22]. The research by Das & Teng (Das and Teng, 2001) [23] also shows the important role of non-official organization mechanisms such as trust governance to strategic alliances performance. As an important governance method in product innovation networks, trust reduces the transaction cost of cooperation among network members (Tortoriello and Krackhardt, 2010) [24]. The development of a trust relationship introduces difficulties that competitors cannot easily imitate, considering the complicated social nature and features of the trust relationship (Dyer and Singh, 1998) [20]. With the deepening of trust, partners will share respective resources and tacit knowledge among each other without worrying about being extorted and betrayed (such as gaining profit by stealing knowledge).

Since trust governance enables alliance partners to feel cared about, it is likely to achieve the objectives of innovation and cooperation, compared with a relationship established and controlled by contract. As the foundation for cooperation, trust is affected by factors such as asset specificity, reputation, capability and substitutability. In a network featuring high levels of trust, enterprises perform transactions, share higher commitments and cognition and tend to show higher levels of interaction (such as knowledge sharing and transfer). Therefore, the feeling of a strong commitment and a high exchange quality brought about by high levels of trust can efficiently mitigate the adverse effects caused by an excessive number of network structure holes, facilitating the exploratory technological innovation of enterprises (Louise, M.M, 2010) [25].

Unlike the compulsory means of contract, trust governance results in a subjective desire on the part of the alliance members for self-restraint. This desire is promoted by flexible means of mutual positive anticipation among cooperating partners and the principle of mutual benefits, thereby effectively preventing moral risks and enhancing the members' willingness to transfer knowledge (Levin and Cross, 2004) [26].

As for the different connotations of exploratory innovation and exploitative innovation, exploratory innovation uses tacit knowledge of high heterogeneity for the development of new products or technologies in new fields. With the increase of inter-enterprise heterogeneity, the difficulty of knowledge transfer is increased. With a high level of trust among alliance members, tacit agreements will be established between members concerning the correct behavior boundary and a good self-implementation mechanism, so that members will conscientiously resist their opportunistic behaviors and trust the partners' motivation for cooperation. In this way, the inter-enterprise heterogeneity and transfer of tacit knowledge will be promoted, thereby facilitating exploratory innovation. In addition to enhancing the willingness to share knowledge among enterprises, non-official governance features, such as trust and mutual benefits, will improve the relative absorption ability of enterprises by enhancing the close interaction among alliance members, encouraging tests of different combinations of knowledge and contributing to jointly solving problems (Dyer and Nobeoka, 2000; Uzzi, 1997) [27,28]. On this basis, Phelps (2010) [10] proposed positively moderating the relationship between network heterogeneity and exploratory innovation based on mutual benefit and trust via a close network.

Based on the above theoretical analysis, we proposed the following hypotheses.

**Hypothesis 3 (H3).** *Trust has a positive moderating effect on the relationship between inter-enterprise knowledge heterogeneity and exploratory innovation performance.*

**Hypothesis 4 (H4).** *Trust has a positive moderating effect on the relationship between inter-enterprise knowledge heterogeneity and exploitative innovation performance.*



*2.3. Moderating Role of Contract*

Owing to the nature of competition as a result of opportunism making members face involuntary losses of knowledge, they refused to exert effort and spend resources to achieve union, which resulted in misinformation and spreading the risks of tacit knowledge challenges (Gulati and Singh, 1998) [29]. The network heterogeneous degree intensified this contradiction, resulting in uniqueness because of the high degree of heterogeneity of tacit knowledge, and thereby increasing the coordination costs (Sampson, 2004) [30]. All these transaction risks will reduce cooperation and knowledge sharing, hindering enterprise innovation efforts (Phelps, 2010) [10].

Contract is a formal means to manage the relationships of enterprises in an alliance innovation network. The behavior of the entities of the innovation network is standardized to reduce the risks of cooperation, usually in an official manner, by signing contracts or agreements.

Contract governance is based on the theory of transaction cost, which instituted formal rules, terms and procedures to improve coordination activities during value creation (Poppo and Zenger, 2002; Hoetker and Mellewigt, 2009; Liu et al., 2009; Bouncken et al., 2016) [31–34]. This was achieved by limiting the personal scope of cooperative partners to reduce opportunism during the value occupation period (Williamson, 1991) [35], and by clarifying the rights and obligations of both parties, to reduce the uncertainty and risk of innovation (Li et al., 2010) [36]. Alliances promote the control of alliance activities to incentivize knowledge sharing, especially when knowledge heterogeneity among members is relatively high.

Appropriate contract governance can influence the process of knowledge sharing, reduce risks and opportunistic behaviors in cooperative innovation activities, and thus promote the enterprise's innovation performance (Bouncken et al., 2016) [34]. Poppo and Zenger (2002) [31] believe that a detailed contract actually contributes to promote a long-term cooperation and the exchange of mutual trust. By stipulating the penalties for breach of contract and the cooperation clause, the contract is beneficial for both parties of the transaction, helping them maintain a long-term relationship of cooperation and resource exchange, as well as reduce opportunistic behaviors. We believe that the level of contract governance affects the level of knowledge exchange and sharing among members of the innovation alliance, which further affects the level of innovation performance of the enterprise.

The alliance governance has different effects on the direct efficiency of inter-enterprise knowledge heterogeneity regarding different types of innovation performance of enterprises (exploratory innovation performance and exploitative innovation performance).

As for exploitative innovation, enterprises often modify their products or develop more types of products by modifying and improving existing processes or technology. These innovation activities rely more on contracts without excessive knowledge heterogeneity between partners or the exchange and sharing of complicated knowledge. Contract governance exerts a strong monitoring, control and punishment of opportunistic behaviors during cooperation via external authoritative constraints, curbing potential cooperation risks and promoting exploitative innovation performance.

However, the formulation of over-complete contracts increased costs and was not conducive to the establishment of deep trust between enterprises. Accordingly, it affected the understanding, transmission and absorption of high-heterogeneous knowledge and had a negative impact on the enterprise's exploratory innovation, which was due to exploratory innovation relying more on the exchange and exploitation of different knowledge and technologies.

Based on the theoretical analysis, we proposed the following hypotheses.

**Hypothesis 5 (H5).** *Contract has a negative moderating effect on the relationship between inter-enterprise knowledge heterogeneity and exploratory innovation performance.*

**Hypothesis 6 (H6).** *Contract has a positive moderating effect on the relationship between inter-enterprise knowledge heterogeneity and exploitative innovation performance.*

To sum up, we obtained the theoretical model of this study, as shown in the Figure 1.

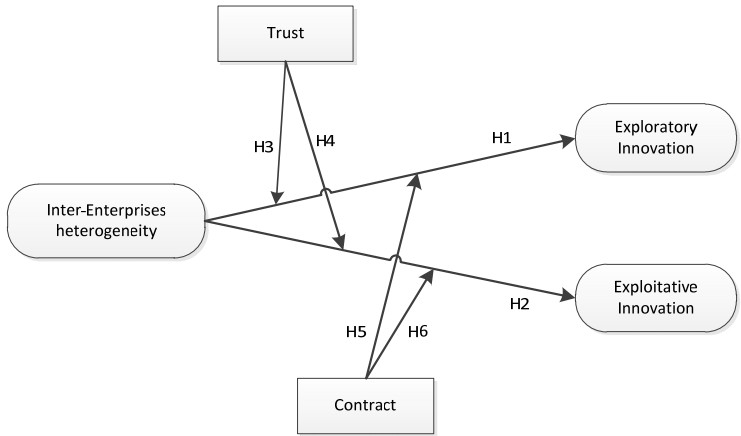

**Figure 1.** The theoretical model.

## 3. Materials and Methods

### 3.1. Procedure and Sample

The selection of the sample of enterprises was mainly based on three factors. First, the enterprises had to engage in product manufacturing activities. Second, they had to conduct product innovation or process innovation activities. Third, they had to have established alliances with multiple innovation partners in recent years. Then, questionnaires were distributed to four developed areas in China, namely, the city of Shanghai and the provinces of Jiangsu, Zhejiang and Guangdong, to reduce the effect of different regional economic development levels. To qualify for our study, participants in those companies needed to be familiar with the company strategy, R&D and overall management. They also needed to serve as managers with more than three years of work experience.

The questionnaires were in paper and electronic forms. We produced the questionnaires by hiring the professional data company Survey Star (http://www.sojump.com, Room 501, Building 7, 1st Phase, Zhongdian Software Park, No.39 Jianshan Road, Yue Lu District, Changsha City, Hunan Province, China), as in other studies in top management journals (e.g., Colquitt, Baer, Long and Halvorsen, 2014; Judge, Simon, Hurst and Kelley, 2013; Rodell and Judge, 2009; Vogel et al., 2016) [37–40]. The questionnaires were distributed in three ways. First, we sent 500 questionnaires through the professional data company Survey Star, obtained 343 responses and selected 208 valid questionnaires. Second, we sent the electronic version or web link to respondents by online tools such as email, WeChat and QQ. A total of 168 questionnaires were sent, 130 were received and 61 valid responses were obtained. Third, 135 paper versions were distributed to EMBA students working in the required industries. A total of 135 questionnaires were received and 53 valid questionnaires were selected. In sum, 803 questionnaires were distributed and 608 were collected. To maximize the statistical power, we adopted a rigorous process of data selection, yielding a final sample size of 322 responses, corresponding to a 75.7% recovery rate and a 52.9% retention rate.

The sample mainly included high-tech industry enterprises such as electronic communications, machinery and instrumentation, bio-pharmaceutical, automobile manufacturing and aerospace. The description of the sample enterprises involved in the study is shown in Table 1.

**Table 1.** Characteristics of the distribution of enterprises.

| Indicator | Industry | Number | Percentage (%) |
|---|---|---|---|
| Industry | Electronic communication | 169 | 52.5% |
| | Machinery and instrumentation | 59 | 18.3% |
| | Bio-pharmaceutical | 32 | 10.0% |
| | Automobile manufacturing and aerospace | 21 | 6.5% |
| | Other industries | 41 | 12.7% |
| Number of employees | <100 | 52 | 16.1% |
| | 101–500 | 167 | 51.9% |
| | 501–1000 | 56 | 17.4% |
| | 1001–5000 | 33 | 10.2% |
| | >5000 | 14 | 4.4% |
| Firm age (years) | <3 | 9 | 2.8% |
| | 3–5 | 19 | 5.9% |
| | 6–10 | 61 | 18.9% |
| | 11–15 | 85 | 26.4% |
| | >15 | 148 | 46.0% |
| Average sales for the last three years (billion RMB) | <0.1 | 53 | 16.5% |
| | 0.1–0.5 | 179 | 55.6% |
| | 0.6–1 | 31 | 9.6% |
| | 1.1–3.0 | 22 | 6.8% |
| | 3.1–6.0 | 10 | 3.1% |
| | 6.1–10.0 | 8 | 2.5% |
| | >10.0 | 19 | 5.9% |
| Ownership of firm | State or state-controlled | 35 | 10.9% |
| | Private | 182 | 56.5% |
| | Wholly foreign-owned | 54 | 16.7% |
| | Joint venture | 51 | 15.8% |

### 3.2. Measures and Methods

We created a Chinese version of all measures by following Brislin's (1976) [41] translation–back-translation procedure. Unless otherwise explained, all variables were measured by participant responses to questions on a seven-point Likert-type scale ranging from "strongly disagree" to "strongly agree." All items of the scale are presented in Appendix A.

**Dependent variable.** We define innovation performance as two dimensions of exploratory innovation performance and exploitative innovation, which refer to the scale created or revised by Jensen (2006) [42]; Lubatkin, Simsek, Ling et al. (2006) [43], Benner and Tushman (2008) [44], Li and Atuahene (2011) [45]. Respondents were required to evaluate the extent of an enterprise's exploratory innovation and utilization of innovation performance compared with the industry average.

**Independent variable.** For the measure of inter-enterprise knowledge heterogeneity, some scholars adopted patent data. However, others believe that this measurement has many drawbacks, such as a lack of implicit technical knowledge indicators (Silverman, 1999) [46], the technology introduction being overlooked, purchase or licensing requirements and the influence of technology classification standards being disregarded (Patel and Pavitt, 1997) [47]. In the Chinese context, the awareness of patent applications (including self-owned patents and cooperation patents) has been strengthened but it is still not very active. Therefore, researchers consider the use of patent application data to measure inter-enterprise knowledge heterogeneity. They easily concluded that the degree of knowledge heterogeneity is generally low, which may not reflect reality. Nieto and Quevedo (2005) [48] emphasize that the questionnaire method provides a broader and accurate view of enterprises' technical knowledge. Therefore, referring to the findings of Granstrand (1998) [49], Rodan and Galunic (2004) [50], Nieto and Quevedo (2005) [48] and Tsai et al. (2014) [51], and to the interviews in this study, we developed a five-item scale to measure inter-enterprise

knowledge heterogeneity. (1) The technical knowledge fields involved are quite different from each other, (2) Considerable differences exist in different technology investment fields, (3) Considerable differences exist in the professional backgrounds of technical staff, (4) Considerable differences exist between production processes, (5) Patent categories are quite different from each other. Respondents were required to evaluate the extent to which the technical knowledge of the enterprise differed from that of the alliance innovation cooperative partners in the past three years.

**Moderating variables.** We used four items developed by Jap and Ganesan (2000) and Lui and Ngo (2004) [52,53] to measure trust and three items developed by Poppo and Zenger (2002) [31] to measure contract. Respondents were required to assess the extent to which the alliance network governance mechanism of their companies operated in the past three years.

**Control variables.** Following existing research practice (Lee and Chen, 2009; Zhou and Li, 2012) [54,55], we selected four control variables, including enterprise size, enterprise age, industry and market environment uncertainty. Since these variables may affect the enterprise network position, inter-enterprise knowledge heterogeneity and innovation performance (Lee and Chen, 2009) [54], we controlled them in the regression model.

We conducted an EFA (exploratory factor analysis) for the five variables. Given that our data are perceptual and collected from a single source at one point in time, we conducted a CFA (confirmatory factor analysis) to test common method bias.

EFA is a method frequently used in scale development. In case of a lack of clear theoretical expectation for the internal structure of the scale, or if related measured indicators are used for the first time—given that it is impossible to exactly judge whether such measured indicators can represent the constructs measured—all indicators are usually measured together, a factor analysis is carried out with regard to their scores and the construct validity is judged based on the factor load value obtained.

CFA is used when researchers already have a clear expectation of the relationship between the constructs and the measured indicators, in order to infer the measured structure by observing the consistency between the tested indicators and the hypothesized model. If no new scales are developed, CFA should be used.

To test our hypotheses, we conducted a hierarchical regression analysis to examine the main effect and moderating effect with the SPSS 19.0 software.

## 4. Results

### 4.1. Validity of Variable Scales

The reliability and validity tests for our measurement items and scales are shown in Appendix A. All the standardized factor loadings in the model were above the commonly accepted value of 0.6 and significantly loaded on their respective factors. The Cronbach's alpha ($\alpha$) values were all above the benchmark value of 0.7. The values of composite reliability (CR) were all above the benchmark value of 0.6. The scores of average variance extracted (AVE) were all above the benchmark value of 0.5. These results indicate that the measurement model has satisfactory convergent validity and reliability.

### 4.2. Descriptive Statistics

The means, standard deviations and correlations among all variables are displayed in Table 2. Inter-enterprise knowledge heterogeneity was positively correlated with exploratory innovation performance ($r = 0.39$, $p < 0.01$) and was significantly and negatively correlated with exploitative innovation performance ($r = 0.20$, $p < 0.01$), suggesting that our hypotheses had preliminary support. The correlation coefficients of variables were lower than 0.7, and the AVE square root value was greater than the correlation coefficient of variables, meaning the model is suitable for further analysis.

**Table 2.** Means, Standard Deviations and Correlations.

| | (1) | (2) | (3) | (4) | (5) | (6) | (7) | (8) |
|---|---|---|---|---|---|---|---|---|
| Exploratory innovation performance | 1.00 | | | | | | | |
| Exploitative innovation performance | 0.48 ** | 1.00 | | | | | | |
| Inter-enterprise knowledge heterogeneity | 0.39 ** | 0.20 ** | 1.00 | | | | | |
| Network centrality | 0.42 ** | 0.34 ** | 0.28 ** | 1.00 | | | | |
| Network intensity | 0.52 ** | 0.58 ** | 0.25 ** | 0.48 ** | 1.00 | | | |
| Network quality | 0.45 ** | 0.49 ** | 0.28 ** | 0.47 ** | 0.53 ** | 1.00 | | |
| Contract | 0.39 ** | 0.47 ** | 0.17 ** | 0.56 ** | 0.50 ** | 0.49 ** | 1.00 | |
| Trust | 0.49 ** | 0.61 ** | 0.17 ** | 0.403 ** | 0.56 ** | 0.42 ** | 0.53 ** | 1.00 |
| Mean | 5.18 | 5.73 | 4.80 | 5.67 | 5.81 | 5.85 | 5.96 | 5.87 |
| Standard deviations | 1.40 | 0.87 | 1.51 | 0.90 | 0.95 | 0.93 | 0.92 | 0.77 |

Note: $N = 322$. ** $p < 0.01$; * $p < 0.05$.

### 4.3. Tests of Hypotheses

To test the hypotheses, we performed hierarchical regression analyses using SPSS 19.0. All interaction variables were mean-centered to minimize multicollinearity (Aiken and West, 1991) [56]. The results of all models are presented in Table 3.

Here is a simple explanation of Models 1–10.

Model 1 added the independent variable inter-enterprise knowledge heterogeneity and the dependent variable exploratory innovation, and tested the main effect of inter-enterprise knowledge heterogeneity on exploratory innovation. Model 2, on the basis of Model 1, added the moderator variable trust. Model 3, on the basis of Model 2, added the product term of the independent variable and trust, and tested the moderating effect of trust on the inter-enterprise knowledge heterogeneity and exploratory innovation. Model 4, on the basis of Model 1, added a moderator variable contract. Model 5, on the basis of Model 4, added the product term of independent variables and contract, and tested the moderating effect of contracts on the inter-enterprise knowledge heterogeneity and exploratory innovation.

Model 6 added the independent variable inter-enterprise knowledge heterogeneity and the dependent variable exploitative innovation, and tested the main effect of inter-enterprise knowledge heterogeneity on exploitative innovation. Model 7, on the basis of Model 6, added the moderator variable trust. Model 8, on the basis of Model 7, added the product term of the independent variable and trust, and tested the moderating effect of trust on inter-enterprise knowledge heterogeneity and exploitative innovation. Model 9, on the basis of Model 6, added a moderator variable contract. Model 10, on the basis of Model 9, added the product term of the independent variable and contract, and tested the moderating effect of contract on inter-enterprise knowledge heterogeneity and exploitative innovation.

As shown in Table 3, the results supported Hypothesis 1—inter-enterprise knowledge heterogeneity is positively related to exploratory innovation performance ($\beta = 0.26$, $p < 0.01$, Model 1). However, Hypothesis 2—inter-enterprise knowledge heterogeneity is negatively related to exploitative innovation performance—was not supported. The regression coefficient of Model 6 showed that inter-enterprise knowledge heterogeneity is positively related to exploitative innovation performance ($\beta = 0.18$, $p < 0.05$, Model 6), which contradicts Hypothesis 2.

**Table 3.** Results of the hierarchical regression analyses.

| | Explorative Innovation Performance | | | | | Exploitative Innovation Performance | | | | |
|---|---|---|---|---|---|---|---|---|---|---|
| | **Model 1** | **Model 2** | **Model 3** | **Model 4** | **Model 5** | **Model 6** | **Model 7** | **Model 8** | **Model 9** | **Model 10** |
| **Control variables** | | | | | | | | | | |
| Market uncertainty | −0.03 ** | −0.03 * | −0.03 * | −0.03 * | −0.03 * | −0.03 ** | −0.04 * | −0.03 * | 0.34 * | 0.35 * |
| Scale | 0.03 | 0.03 | 0.02 | 0.05 | 0.06 | 0.06 | 0.04 | 0.04 | 0.04 | 0.04 |
| Age | 0.12 | 0.12 | 0.08 | 0.09 | 0.11 | 0.11 | 0.09 | 0.09 | 0.09 | 0.09 |
| Industry | 0.07 | 0.07 | 0.05 | 0.06 | 0.08 | 0.06 | 0.05 | 0.04 | 0.04 | 0.04 |
| **Independence** | | | | | | | | | | |
| Inter-enterprise knowledge heterogeneity | 0.26 ** | 0.23 ** | 0.22 ** | 0.19 ** | 0.18 ** | 0.18 * | 0.17 * | 0.20 * | 0.18 * | 0.16 * |
| **Mediating variables** | | | | | | | | | | |
| Trust | | 0.15 * | 0.13 * | | | | 0.14 ** | 0.12 * | | |
| Contract | | | | 0.12 * | 0.14 | | | | 0.16 ** | 0.17 ** |
| **Cross terms** | | | | | | | | | | |
| Inter-enterprise knowledge heterogeneity × trust | | | 0.18 ** | | | | | 0.13 * | | |
| Inter-enterprise knowledge heterogeneity × contract | | | | | −0.15 * | | | | | 0.14 ** |
| $R^2$ | 0.47 | 0.48 | 0.48 | 0.50 | 0.48 | 0.48 | 0.52 | 0.58 | 0.61 | 0.60 |
| Adjusted $R^2$ | 0.45 | 0.45 | 0.47 | 0.49 | 0.47 | 0.47 | 0.50 | 0.57 | 0.59 | 0.58 |
| F | 37.0 ** | 37.0 ** | 29.3 ** | 31.9 ** | 33.5 ** | 33.5 ** | 32.3 ** | 35.2 ** | 29.9 ** | 28.8 ** |

Note: ** $p < 0.01$; * $p < 0.05$.

We predicted that trust positively moderates the relationship between inter-enterprise knowledge heterogeneity and exploratory innovation performance (Hypothesis 3) and that trust positively moderates the relationship between inter-enterprise knowledge heterogeneity and exploitative innovation performance (Hypothesis 4). All interaction variables were mean-centered to minimize multicollinearity (Aiken and West, 1991) [49]. As shown in Table 3, the interaction between trust and inter-enterprise knowledge heterogeneity was positively related to exploratory innovation performance ($\beta$ = 0.18, $p$ < 0.05, Model 3), and the interaction between trust and inter-enterprise knowledge heterogeneity was positively related to exploitative innovation performance ($\beta$ = 0.13, $p$ < 0.05, Model 8). The change of the multiple squared correlation coefficient ($\Delta R^2$) for the interaction term of trust and inter-enterprise knowledge heterogeneity was statistically significant, explaining a significant amount of variance in explorative innovation performance ($\Delta R^2 = 0.01, p < 0.05$). $\Delta R^2$ for the interaction term of trust and inter-enterprise knowledge heterogeneity was statistically significant, explaining a significant amount of variance in exploitative innovation performance ($\Delta R^2 = 0.1, p < 0.05$). To determine the nature of the moderating effect, we plotted the slope tendency of the regression equations for the connection of inter-enterprise knowledge heterogeneity and exploratory innovation performance and the connection of inter-enterprise knowledge heterogeneity and exploitative innovation performance, according to the high and low values of trust. Following the method of Cohen (1983) [57], we defined the high and low values as plus and minus one standard deviations from the mean, to obtain the figures. Figure 2 shows that the interaction pattern, as predicted in the relationship between inter-enterprise knowledge heterogeneity and exploratory innovation performance, was stronger for enterprises with high trust. Figure 3 shows that the interaction pattern, as predicted in the relationship between inter-enterprise knowledge heterogeneity and exploitative innovation performance, was stronger for enterprises with high trust.

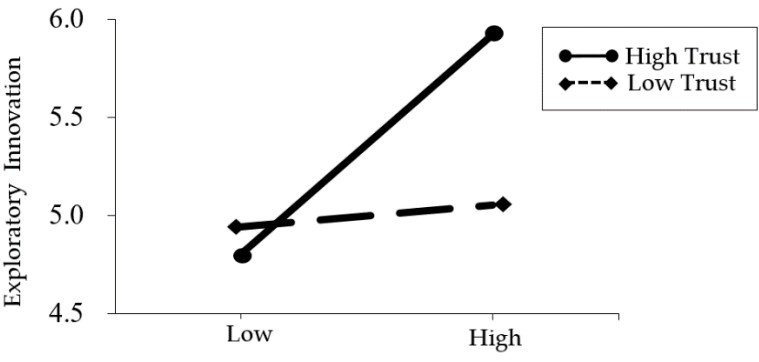

**Figure 2.** The interactive effect of inter-enterprise knowledge heterogeneity and trust on exploratory innovation.

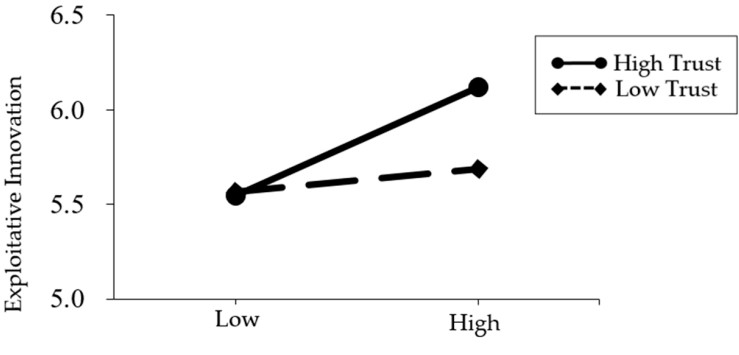

**Figure 3.** The interactive effect of inter-enterprise knowledge heterogeneity and trust on exploitative innovation.

We predicted that contract negatively moderates the relationship between inter-enterprise knowledge heterogeneity and exploratory innovation performance (Hypothesis 5) and that contract positively moderates the relationship between inter-enterprise knowledge heterogeneity and exploitative innovation performance (Hypothesis 6). All interaction variables were mean-centered to minimize multicollinearity (Aiken and West, 1991) [49]. As shown in Table 3, the interaction between contract and inter-enterprise knowledge heterogeneity was negatively related to exploratory innovation performance ($\beta = -0.15$, $p < 0.05$, Model 5), and the interaction between contract and inter-enterprise knowledge heterogeneity was positively related to exploitative innovation performance ($\beta = 0.14$, $p < 0.05$, Model 10). The value of $\Delta R^2$ for the interaction term of contract and inter-enterprise knowledge heterogeneity was statistically significant, explaining a significant amount of variance in explorative innovation performance ($\Delta R^2 = 0.03$, $p < 0.05$). $\Delta R^2$ for the interaction term of contract and inter-enterprise knowledge heterogeneity was statistically significant, explaining a significant amount of variance in exploitative innovation performance ($\Delta R^2 = 0.12$, $p < 0.05$). To determine the nature of the moderating effect, we plotted the slope tendency of the regression equations for the connection of inter-enterprise knowledge heterogeneity and exploratory innovation performance and the connection of inter-enterprise knowledge heterogeneity and exploitative innovation performance, according to the high and low values of contract. Figure 4 shows that the interaction pattern, as predicted in the relationship between inter-enterprise knowledge heterogeneity and exploratory innovation performance, was stronger for enterprises with high contract ("high contract" refers to a high level of contract as moderator variable, which describes the tendency of enterprises to adapt the contract method in more complete, clear, detailed and comprehensive terms when governing alliance enterprises). Figure 5 shows that the interaction pattern, as predicted in the relationship between inter-enterprise knowledge heterogeneity and exploitative innovation performance, was stronger for enterprises with high contract.

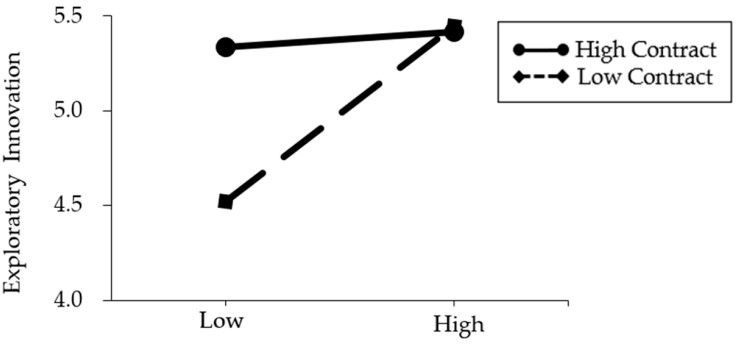

**Figure 4.** The interactive effect of inter-enterprise knowledge heterogeneity and contract on exploratory innovation.

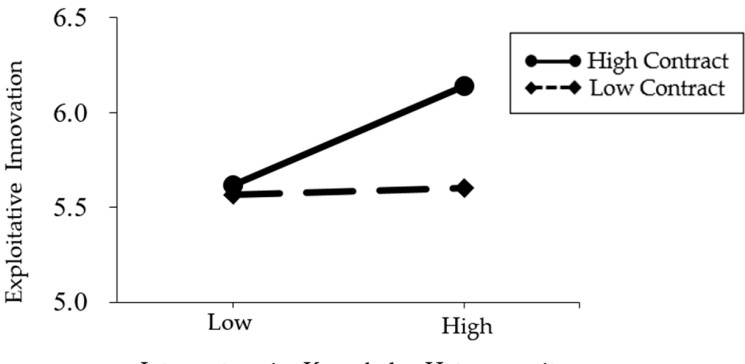

**Figure 5.** The interactive effect of inter-enterprise knowledge heterogeneity and contract on exploitative innovation.

## 5. Discussion and Conclusions

*5.1. Theoretical Implications*

Some conclusions of this study are supported by empirical data. Inter-enterprise knowledge heterogeneity is positively correlated with both exploitative and exploratory innovation performance in a significant way. Exploratory innovation requires a large reserve of heterogeneous knowledge from alliance partners, and an increase in inter-enterprise knowledge heterogeneity is conducive to the rapid and extensive acquisition of new technologies and capabilities, to the research and development of brand new products and processes and thus to the enhancement of exploratory innovation performance. We believe that inter-enterprise knowledge heterogeneity provides enterprises with more diverse knowledge and information, different from their own areas of knowledge, and exploratory innovation is precisely a technological innovation activity that needs to acquire and create brand new knowledge and that strives to break away from, and move beyond, the existing knowledge base. Thus, inter-enterprise knowledge heterogeneity is able to boost exploratory technological innovation performance.

Exploitative innovation aims to enhance existing technologies or skills or improve existing products and processes of enterprises in response to market demands, with a focus on the integration of internal knowledge resources. However, attention must also be paid to the introduction of external heterogeneous knowledge, which shapes the mode of thinking and the effects of knowledge integration, thereby affecting the performance of exploitative innovation.

The current literature regarding the relationship between knowledge heterogeneity and innovation performance presents different or even opposite conclusions [1–7,50,58]. Considering that these studies did not research the dimensions of innovation performance (exploitative or exploratory) separately, this paper contributes to the existing research by drawing on the studies with contradictory conclusions to discuss the relationship between knowledge heterogeneity and innovation performance from the perspectives of exploitative innovation and exploratory innovation.

Contrary to the original hypothesis, a positive correlation is indicated in this paper between knowledge heterogeneity and exploratory innovation performance. Theoretically, the acquisition of external heterogeneous knowledge expands the basic corporate knowledge reserve, improves the current products and technology and enhances exploitative innovation performance. Judging by their practical situation, the high-tech companies of the same industry in China generally lag behind in inter-enterprise knowledge heterogeneity, meaning that the inter-enterprise knowledge heterogeneity of these companies has not reached the level where the understanding, absorption, integration and utilization of external tacit knowledge become difficult. Accordingly, the current absorption capacity of companies allows them to utilize heterogeneous knowledge from other companies to promote their own exploitative innovation. Therefore, inter-enterprise knowledge heterogeneity has a positive effect on exploitative innovation, and not the negative correlation assumed at the beginning of this study.

Alliance governance mechanisms have varying moderating effects on the relationship between inter-enterprise knowledge heterogeneity and innovation performance. Specifically, trust and contract have moderating effects on exploratory innovation and exploitative innovation. Trust has a positive moderating effect on the relationship between inter-enterprise knowledge heterogeneity and exploratory innovation performance. Moreover, trust also has a positive moderating effect on the relationship between inter-enterprise knowledge heterogeneity and exploitative innovation performance. Contract has a negative moderating effect on the relationship between inter-enterprise knowledge heterogeneity and exploratory innovation performance. Moreover, contract also has a positive moderating effect on the relationship between inter-enterprise knowledge heterogeneity and exploitative innovation performance.

The effects of alliance governance mechanisms vary depending on the type of innovation performance (exploratory and exploitative). Contract mainly exerts a powerful supervision, control and punishment of opportunistic behavior during cooperation through the binding force of external authorities, curbing potential cooperation risks and boosting innovation performance. Unlike coercive

contract, trust resorts to flexible means such as the positive mutual expectations of partners and the principles of reciprocity and mutual benefits for alliance members, contributing to a subjective desire for self-discipline, preventing moral hazards in cooperation and enhancing the willingness of alliance members to transfer knowledge (Levin and Cross, 2004) [26].

As for the different connotations of exploratory and exploitative innovation, exploratory innovation mainly uses highly heterogeneous, tacit knowledge for the development of new products or technologies in new fields, and an increase in inter-enterprise knowledge heterogeneity will add to the difficulty of knowledge transfer. If a high degree of trust exists among alliance members, a tacit understanding of each other's behavior boundaries is established, and a sound self-execution mechanism is formed. Thus, alliance members will consciously resist their own opportunistic behavior and trust each other's motives for cooperation, which will effectively promote inter-enterprise heterogeneous and tacit knowledge transfer, and thus facilitate exploratory innovation.

A contract that is too sound means a higher perceived risk of opportunistic behavior, that needs to be controlled by many contractual means and will possibly undermine the foundation of trust between the partners and cause negative emotions of distrust. In response, the wronged partner often adopts non-cooperative, "tit for tat" strategies, including deliberately restricting or suspending the heterogeneous knowledge sharing and transfer that is necessary to the cooperative innovation process, eventually affecting exploratory innovation performance.

Exploitative innovation mainly involves the improvement of existing products, the development of more product types or the modification and enhancement of existing processes or technologies. Such innovation activities do not entail the high-level exchange and sharing of heterogeneous or complicated knowledge among alliance partners but rely on the performance of contractual relationships. In these cases, a high degree of trust should exist among alliance partners, and a reduction in free-riding or inaction will occur, resulting in complementary effects of contract and trust mechanisms on exploitative innovation.

## 5.2. Practical Implications

This study has a certain importance in management practice. It may help entrepreneurs contingently set alliance governance mechanisms to effectively govern alliance partners and maximize the effect of inter-enterprise heterogeneous knowledge on different types of innovation performance. In alliance cooperative innovation, the governance of alliance partners is crucial. Given that the partners of alliance cooperation are diverse in nature and function, the target enterprise may implement different governance methods or a combination of methods for different alliance partners and even establish a special alliance governance mechanism for management. From entering into a contract to establishing trust, this process is one of continuous reinforcement of alliance governance mechanisms. When the target enterprise sets governance mechanisms properly, it will enable alliance partners to establish a common vision. This vision entails the willingness to share risks, a conscious self-supervision and a low risk of opportunism. Such a contingent setting of combined governance mechanisms will lead to open resources, proprietary skills and tacit knowledge among alliance members, without fear of blackmail, betrayal or theft. The joint governance of contract and trust mechanisms will play an active role in corporate governance and alliance partnerships. Enterprise managers may set governance mechanisms contingently, according to the type of enterprise innovation (exploratory or exploitative), to effectively boost enterprise innovation performance. For example, they can set high trust–low contract or low trust–high contract complementary governance mechanisms to govern different alliance partnerships, realize the objectives of cooperative innovation and achieve the continued development of innovation performance.

## 5.3. Future Directions and Limitations

This study mainly adopted the questionnaire survey and case study methods for empirical analysis. Despite the large size of the survey sample, the subjects are concentrated in the economically developed

regions of Jiangsu, Zhejiang, Shanghai and Guangdong, in China, which means the study is not universal. Despite the relatively satisfactory reliability and validity results of the measurement of inter-enterprise knowledge heterogeneity, its scientificity and rationality must be further verified. In addition, this study only introduced trust and contract in alliance governance mechanisms as moderating variables in the relationship between inter-enterprise knowledge heterogeneity and innovation performance, but formal and informal alliance governance mechanisms also include reciprocity governance and learning governance. Future research may explore how these factors affect the relationship between inter-enterprise knowledge heterogeneity and innovation performance.

In addition, the study also discussed the relationship between the knowledge heterogeneity and innovation performance of enterprises, taking into account the characteristics (types of ownership structure) of hi-tech enterprises in China, the competency of core enterprises in the innovation network and the relationship with cooperating innovation members, in order to find the reasons for the existing differences, as well as their process and results.

**Author Contributions:** C.T. and Y.Q. conceived and designed the study. C.T. and Z.G. designed the questionnaire and collected the data. C.T. and Y.Q. conducted the data analysis. C.T. and Y.Q. wrote the manuscript, and H.R. supervised and reviewed the manuscript. All authors have read and agreed to the published version of the manuscript.

**Funding:** This research was funded by Science Foundation of the Ministry of Education of China (19C10276025).

**Conflicts of Interest:** The authors declare no conflict of interest.

# Appendix A

**Table A1.** Scales for measuring variables and validity of all the variables.

| Items | CITC | Factor Loading | Validity |
|---|---|---|---|
| Exploratory Innovation ($\alpha$ = 0.889; CR = 0.859; AVE = 0.670) | | | $\chi2$ = 544.007; Df = 3; |
| (1) I often try to use new technologies/skills that are not yet mature and have certain risks | 0.777 | 0.813 | $p < 0.01$; |
| (2) I often try to develop technologies in new areas | 0.784 | 0.820 | CFI = 0.985; TLI = 0.975; IFI = 0.985; |
| (3) I frequently try business strategies/tactics that have not been used by other companies in the same industry | 0.787 | 0.822 | RMSEA = 0.072 |
| Exploitative Innovation ($\alpha$ = 0.842; CR = 0.894; AVE = 0.678) | | | $\chi2$ = 543.875; Df = 6; |
| (1) I constantly improve existing technologies/skills to adapt to market needs | 0.696 | 0.838 | $p < 0.01$; |
| (2) I strive to improve the applicability of existing technologies/skills in multiple relevant business areas | 0.648 | 0.803 | CFI = 0.985; TLI = 0.975; IFI = 0.985; |
| (3) I frequently use existing technologies/skills to add function and variety to products/services | 0.700 | 0.840 | RMSEA = 0.072 |
| (4) I frequently refine the business experience accumulated by the company and apply it to the current business | 0.660 | 0.813 | |
| Inter-enterprise Knowledge Heterogeneity ($\alpha$ = 0.925; CR = 0.880; AVE = 0.610) | | | $\chi2$ = 1203.182; Df = 10; |
| (1) The technical knowledge fields involved are quite different from each other | 0.807 | 0.774 | $p < 0.01$; |
| (2) Considerable differences exist in different technology investment fields | 0.794 | 0.758 | CFI = 0.995; TLI = 0.994; |
| (3) Considerable differences exist in the professional backgrounds of technical staff | 0.779 | 0.737 | IFI = 0.996; |
| (4) Considerable differences exist between production processes | 0.812 | 0.778 | RMSEA = 0.032 |
| (5) Patent categories are quite different from each other | 0.833 | 0.806 | |
| Trust ($\alpha$ = 0.831; CR = 0.887; AVE = 0.66) | | | $\chi2$ = 466.615; Df = 6; |
| (1) Our partners can always be trusted | 0.630 | 0.834 | $p < 0.01$; |
| (2) Our partners always keep their promises | 0.619 | 0.799 | CFI = 0.94; TLI = 0.92; |
| (3) We believe in the ability of our partners | 0.687 | 0.836 | IFI = 0.94; |
| (4) Our partners are working hard to complete tasks without monitoring | 0.624 | 0.788 | RMSEA = 0.078 |
| Contract ($\alpha$ = 0.864; CR = 0.846; AVE = 0.647) | | | $\chi2$ = 458.125; Df = 3; |
| (1) Detailed contracts are the best way to ensure the success of collaborative innovation | 0.754 | 0.800 | $p < 0.01$; |
| (2) Contracts are the best way to manage cooperative innovation enterprises | 0.758 | 0.805 | CFI = 0.94; TLI = 0.92; IFI = 0.94; |
| (3) Both sides in the cooperation want to include all the details in the contract | 0.711 | 0.755 | RMSEA = 0.078 |
| Market Uncertainty ($\alpha$ = 0.713; CR = 0.840; AVE = 0.639) | | | $\chi2$ = 183.494; Df = 3; |
| (1) Customers raise new requirements for products and services constantly | 0.514 | 0.786 | $p < 0.01$; CFI = 0.94; |
| (2) The market environment is constantly changing | 0.572 | 0.830 | TLI = 0.98; IFI = 0.98; |
| (3) Customer requirements on the quantity and delivery time of products/services change frequently | 0.508 | 0.775 | RMSEA = 0.08 |

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
