# Peer review of "The Influence of Inter-Enterprise Knowledge Heterogeneity on Exploratory and Exploitative Innovation Performance: The Moderating Role of Trust and Contract"

_sustainability, doi:10.3390/su12145677_

Round 1
Reviewer 1 Report
The study explores the impact of knowledge heterogeneity on exploratory and exploitative innovation in Chinese firms and the moderating role of trust and contractual arrangements. The following comments should be addressed to improve the study.
What is the theoretical background of the study? Is it e.g. the resource-based view of the firm, absorptive capacity, the open innovation framework, etc.?
What are the differences and similarities between exploratory and exploitative innovations on the one hand and radical and incremental innovations on the other hand?
The discussion on the impact of contracts on innovation is missing references (page 4, lines 181-190). By signing contracts with their collaborative partners, firms are protected from their opportunistic behaviour. Also, in terms of exploratory (radical) innovations, contracts stipulate which partner will have the Intellectual property rights (IPRs), in case of patent(s) being invented during cooperation. Accordingly, if we argue that exploratory (radical) innovation is more likely to results in patents than exploitative (incremental) innovation, we can argue that contracts have a positive effect on exploratory (radical) innovations.
Section 4.2. should be descriptive statistics, not description statistics.
What is meant by high contract (line 337)?
In lines 368-374, it is stated: “Existing literature mostly holds that knowledge heterogeneity plays a role in promoting innovation performance and that only inter-enterprise knowledge heterogeneity beyond a critical point might negatively affect exploitative innovation, but inter-enterprise knowledge heterogeneity of the domestic respondents in the alliance network may not reach the theoretical critical point. Therefore, inter-enterprise knowledge heterogeneity has a positive effect on exploitative innovation, not a negative correlation assumed at the beginning of this study.” The author(s) do not test a potential tipping (critical) point, as the squared term is required. So I don’t understand how this statement can be associated with the current study. Also the statement “only inter-enterprise knowledge heterogeneity beyond a critical point might negatively affect exploitative innovation” means that knowledge heterogeneity has a positive effect up to the tipping point. So why would author(s) assume that firms in the study have reached beyond the tipping point (when negative effect occur)?
Why aren't both variables trust and contract included in all 10 models in Table 4?
Reviewer 2 Report
Dear Sir/ Madam
I find the article extremely interesting and worth publishing. The research is well grounded in the theory layer. it also depicts its practical aspects in managing the innovation processes in enterprises. I would like to encourage the Authors to continue their work connected with the surveyed business entities taking into consideration the possibility of using the taxonomic structure to depict similarities and differences in this respect taking into consideration the size classes of the examined units, the kind of their business activity, the period of their existence on the market or their ownership form. At present the research depicts rather differentiated business units which in turn not necessarily accounts for the correctness of the methodical approach presented in such a broad perspective (the accepted determinant is the membership in the High Tech group of enterprises). The article nevertheless possesses its added value, but on the other hand my suggestion is to state whether the defined differences between the enterprises become visible taking into consideration the criteria which were enumerated before (class size, kind of business activity etc.) In my opinion it is extremely desirable.
Other remarks:
1) In table 1 please state the currency (average sales for the last 3 years in billion).
2) In table 4 it is written: Model 1 – model 9, but the last one is model 0. Is this correct?
3) In the article text there is no information (description) on what presents Model 1, 2… This information ought to be placed under the table (as an explanation).
Reviewer 3 Report
As a general comment I can congratulate the authors on the basis of scope. The proposed research topic is in the focus of researchers and practitioners as well.
However there are some minor recommendations from my point of view, which can improve the quality of this research work :
-it might be useful and very interesting to present the selected enterprises from European point of view as well. The percentage of SMEs and large enterprises, the number of enterprises having employee less than 250 and if are micro-enterprises between them (less than 10 employee).
-In the Introduction it might be useful to present the structure of the paper and to underline where and in which section(s) the own contributions are presented.
- it might be useful to move some Tables and Figures to Annex for better understanding of the content, to follow more easier the content (Table 1, 2 and Table 4 for example and Figure 2 -5)
-Line 229 it might be useful to verify the formulation”were presented” or “are presented?”
-Line 241-242 about the patent application data – it is about the enterprise own patent or not?
-Line 247 it might be useful to shortly present the developed five-item scale
-Line 259-260 it might be useful to introduce the mean of EFA and CFA
-Line 274 the correlation under 0,5 it is not significanly correlated, it might be better to write only positively correlated
-Table 4, Line 286 it might be useful to explaine shortly Model-1-9 and Model 10 which in Table is named Model 0
-Line 332 it might be useful to explaine shortly how the Figures were obtained
Round 2
Reviewer 1 Report
The authors have addressed my comments at a satisfactory level.